# Isolated Sagittal Craniosynostosis: A Comprehensive Review

**DOI:** 10.3390/diagnostics14040435

**Published:** 2024-02-16

**Authors:** Peter Spazzapan, Tomaz Velnar

**Affiliations:** 1Department of Neurosurgery, University Medical Centre Ljubljana, 1000 Ljubljana, Slovenia; spazzapanpeter@yahoo.it; 2Alma Mater Europaea ECM, 2000 Maribor, Slovenia

**Keywords:** craniosynostosis, scaphocephaly, cranial deformation, surgery, cranial vault remodelling

## Abstract

Sagittal craniosynostosis, a rare but fascinating craniofacial anomaly, presents a unique challenge for both diagnosis and treatment. This condition involves premature fusion of the sagittal suture, which alters the normal growth pattern of the skull and can affect neurological development. Sagittal craniosynostosis is characterised by a pronounced head shape, often referred to as scaphocephaly. Asymmetry of the face and head, protrusion of the fontanel, and increased intracranial pressure are common clinical manifestations. Early recognition of these features is crucial for early intervention, and understanding the aetiology is, therefore, essential. Although the exact cause remains unclear, genetic factors are thought to play an important role. Mutations in genes such as FGFR2 and FGFR3, which disrupt the normal development of the skull, are suspected. Environmental factors and various insults during pregnancy can also contribute to the occurrence of the disease. An accurate diagnosis is crucial for treatment. Imaging studies such as ultrasound, computed tomography, magnetic resonance imaging, and three-dimensional reconstructions play a crucial role in visualising the prematurely fused sagittal suture. Clinicians also rely on a physical examination and medical history to confirm the diagnosis. Early detection allows for quick intervention and better treatment outcomes. The treatment of sagittal craniosynostosis requires a multidisciplinary approach that includes neurosurgery, craniofacial surgery, and paediatric care. Traditional treatment consists of an open reconstruction of the cranial vault, where the fused suture is surgically released to allow normal growth of the skull. However, advances in minimally invasive techniques, such as endoscopic strip craniectomy, are becoming increasingly popular due to their lower morbidity and shorter recovery times. This review aims to provide a comprehensive overview of sagittal craniosynostosis, highlighting the aetiology, clinical presentation, diagnostic methods, and current treatment options.

## 1. Introduction

The skull of vertebrates consists of the neurocranium, which surrounds and protects the brain, and the viscerocranium, which forms the face. The neurocranium is separated into the following:Membranous neurocranium, which forms through the process of intramembranous ossification and creates the frontal bone, the squamous portion of the temporal bone, the intraparietal portion of the occipital bone, and the parietal bone;Cartilaginous neurocranium, which ossifies via endochondral ossification and gives rise to the ethmoid and sphenoid bones, as well as the petrous and mastoid portions of the temporal bone and the occipital bone. 

The skull sutures form at the junctions of the skull bones. Here the bone tissue is gradually deposited; therefore, the cranial sutures act as ossification centres of the skull [1]. The skull sutures allow unrestricted brain growth through progressive bone deposition. Craniosynostosis occurs when the cranial sutures undergo too early ossification and no longer function as ossification centres. In craniosynostosis, the growth of the skull is impaired due to the ossification of one or more cranial sutures, thus giving rise to specific skull deformations. Virchow’s law [2] states that if the growth of the skull stops in a direction perpendicular to the ossified suture, the rapidly growing brain finds space for its rapid growth elsewhere, and compensatory growth occurs in a direction parallel to the affected suture and no longer in a perpendicular direction. The clinical picture varies depending on which suture is fused [3].

Craniosynostosis is defined as syndromic or non-syndromic. Of the non-syndromic ones, scaphocephaly, trigonocephaly, and anterior plagiocephaly are the most common. Brachycephaly and posterior plagiocephaly are less common. Syndromic craniosynostoses are more complex conditions, usually involving multiple sutures and usually associated with intracranial hypertension, hydrocephalus, and Chiari malformation. Syndromic forms of craniosynostosis usually present with multiple fused sutures and may include central nervous system, limb, or airway malformations [4].

According to literature data, the incidence is estimated at 1/2100 births, with a male-to-female ratio of 4 to 1. Non-syndromic craniosynostosis accounts for 85 to 95% of all cases, while syndromic cases account for 5 to 15%. In Slovenia, the annual incidence is estimated at 1:1500 births. The prevalence of non-syndromic craniosynostosis is 93%, and that of syndromic craniosynostosis is 7%. The predominance of male children (78.9%) is well known [1,3,4,5]. 

In terms of morphological phenotypes, the most common form of craniosynostosis is non-syndromic craniosynostosis (ISS) (40 to 55%) (Figure 1). In second place is the metopic synostosis (15 to 30%), followed by the unilateral and bilateral coronal synostosis (15 to 20%). Lambdoid synostosis is the rarest (0 to 5%). Ossification of two or more sutures is rare and usually occurs in syndromic cases [3]. In Slovenia, ISS accounted for 54.9% of cases, metopic craniosynostosis 25.3%, unilateral coronal craniosynostosis 14%, bilateral coronal craniosynostosis 1.4%, and lambdoid craniosynostosis 1.4%. Multiple suture craniosynostosis occurred in 2.8% of cases [1,5].

Among the known aetiologies of craniosynostosis, genetic, metabolic, and haematological diseases, mucopolysaccharidosis, and teratogens (valproic acid, retinoic acid) are the most important [3]. The presence of CSF drainage and microcephaly can also be a cause of craniosynostosis, as in both cases, the growth potential of the brain itself is reduced; the cranial sutures are not exposed to the force of the growing brain and, consequently, ossify prematurely. Genetic studies have identified some of the genes involved in the pathogenesis of craniosynostosis. The genes FGFR−1, FGFR−2, FGFR−3, FGFR−4, and TGFBR−2 (associated with Loeys–Dietz syndrome) encode receptors that are involved in the processes of differentiation, proliferation, and migration of cranial suture cells. 

The genetic causes of syndromic craniosynostosis also include alterations in the TWIST1, ERF, and EFNB1 genes, which play a specific role in the development of this condition. Understanding the roles of these genes in syndromic craniosynostosis is crucial for diagnosis, genetic counselling, and potentially developing targeted therapies since early identification of mutations in these genes can aid in personalised management and treatment plans for affected individuals [3]. For example, mutations in the TWIST1 gene are associated with Saethre–Chotzen syndrome, one of the syndromes causing craniosynostosis. TWIST1 is involved in the regulation of embryonic development, particularly in the formation of bones and other tissues. Mutations in this gene disrupt normal cranial suture development, leading to premature fusion of the skull bones. ERF (ETS2 repressor factor) is a transcription factor that regulates gene expression by binding to specific DNA sequences. Mutations in ERF have been linked to craniosynostosis, particularly the Muenke syndrome subtype. ERF likely plays a role in controlling the balance of cell proliferation and differentiation during cranial bone development. Disruptions in ERF function can lead to abnormal skull growth and premature fusion of cranial sutures. EFNB1 (Ephrin−B1) is a gene that encodes a cell surface protein involved in cell signaling. Mutations in EFNB1 are associated with craniofrontonasal syndrome, another type of syndromic craniosynostosis. EFNB1 is important for the guidance of migrating neural crest cells during embryonic development, which contribute to the formation of the skull and face. Mutations in EFNB1 disrupt normal cranial bone development, leading to craniosynostosis and other craniofacial abnormalities [1,5].

The diagnosis of craniosynostosis is initially clinical. Depending on the affected suture, characteristic deformations of the cranial vault occur. Other clinical signs are the typical ridge that appears over the ossified suture and the absence of displacement of the two bones adjacent to the suture at palpation. The diagnosis is ultimately confirmed radiologically. Ultrasonography (US) of the head alone confirms the ossification of the suture (Figure 2), and computed tomography (CT) of the head with 3D reconstruction clearly shows the craniosynostosis.

These two examinations are particularly important in those rare cases in which craniosynostosis cannot be confirmed with certainty by clinical examination. An ocular fundus examination is indicated to rule out the presence of increased intracranial pressure. More detailed information about the anatomy of the brain can be obtained by magnetic resonance imaging (MRI), although this is rarely indicated in isolated craniosynostosis, as clinically significant brain abnormalities in craniosynostosis are rare and can usually be recognised by ultrasound of the head. Genetic tests are always necessary if an underlying syndromic condition is suspected. In syndromic forms, however, MRI imaging is frequently used to evaluate associated Chiari malformation, syrinx, hydrocephalus, and other CNS malformations [4,5]. 

The treatment of craniosynostosis is surgical and requires complex remodelling of the deformed parts of the skull [6]. The procedure must eliminate the constricting and deforming tendency that synostosis exerts on the growth of the head. This requires remodelling of the bony structures of the neurocranium and, if necessary, of the viscerocranium. The procedure must be performed not only for cosmetic reasons but, above all, to allow the brain to grow normally. The age at which surgical treatment of craniosynostosis is performed depends on the involved suture, the possible presence of hydrocephalus or Chiari malformation, and the systemic, especially respiratory, condition of the child.

When planning the treatment of a child with craniosynostosis, it is important to consider which suture is affected, the general and neurological condition of the child, and any associated pathologies and malformations. In the vast majority of non-syndromic craniosynostosis, a single procedure is sufficient to reshape the skull. Hydrocephalus or Chiari malformations are very rare in these cases. In syndromic cases, Chiari malformation, raised intracranial pressure, and hydrocephalus are more common [4]. Apneas are also common and occur both due to central reasons (in association with Chiari malformation) and peripherical reasons (in association with maxillary hypoplasia and upper airway stenosis). These are, therefore, complex clinical conditions that need to be corrected by a series of interventions in the correct chronological order. The treatment of such children is time-consuming and multidisciplinary. Surgical treatment of craniosynostosis is complex and not without complications. According to the literature, complications occur in 2 to 8% of cases [7]. The mortality rate in all current series is less than 1%. Possible complications include wound dehiscence, infections, subcutaneous haematomas, dural injuries, and CSF leakage.

The evaluation of the success of craniosynostosis surgery is based on the assessment of the aesthetic and functional outcomes. The aesthetic result is difficult to assess objectively. The most useful methods for this purpose are craniometric measurements, the most important of which is the cranial index (CI), which indicates the ratio of skull width to skull length. In healthy, normocephalic children, the CI is between 76% and 78% [8]. The closer the CI approaches these values after surgery, the more favourable the aesthetic results.

The cognitive outcome is assessed by neuropsychological tests performed throughout the child’s development. In children without concomitant hydrocephalus and without raised intracranial pressure, certain neurocognitive problems occur in 30 to 50% of cases despite surgery [9]. These problems are most pronounced in the areas of language acquisition, writing, and reading, although the intelligence quotient (IQ) is usually within normal limits. In this review, we discuss ISS and its management.

## 2. Isolated Sagittal Craniosynostosis

Scaphocephaly is a skull deformity caused by ISS, i.e., premature, non-syndromic ossification of the sagittal suture. ISS is the most common form of craniosynostosis occurring in clinical practise (incidence 1:5000 births) and accounts for 40 to 60% of all cases [3,10]. It occurs more frequently in boys than in girls. In the neonatal period, it can be easily recognised and diagnosed by a simple clinical examination. It presents with an elongated and narrowed head shape, characterised mainly by frontal and occipital bossing and a narrow bitemporal and biparietal distance (Figure 3). There is no asymmetry of the face, orbits, or brain base in ISS. A thickening of the sagittal suture is often observed, which can be palpated. Radiologically, the thickened suture can be confirmed by a simple X−ray of the head or, more precisely, by CT (Figure 1) or ultrasound (Figure 2).

Although these general data suggest that ISS is a specific, homogeneous, and well-defined pathology, a close examination of each case and a thorough review of the literature may show that this is not the case. In fact, during the routine clinical and radiological examination of a child with ISS, the various anatomical, genetic, and functional abnormalities that can accompany almost every single case of ISS are easily overlooked and not evaluated. When properly recognised, ISS can be considered a much more heterogeneous pathology, which should be evaluated by the multidisciplinary expertise of paediatricians, geneticists, anatomists, neurologists, radiologists, and paediatric neurosurgeons [11].

The sagittal suture does not ossify according to the all-or-nothing principle. Therefore, different morphological forms of scaphocephaly and different clinical presentations may develop depending on the different segments of the sagittal suture involved in the process of craniosynostosis [8,11]:−Dolichocephaly occurs when the entire sagittal suture is ossified. It is characterised by an elongated and narrow head.−Leptocephaly occurs when the anterior third of the suture is ossified. It is characterised by a uniform and homogeneous narrowing of the cranial vault, affecting both the parietal and frontal bones.−Batrocephaly occurs when the middle and anterior third of the sagittal suture are ossified. It is characterised by pronounced occipital bossing.−Cynocephaly occurs when the middle third of the sagittal suture is ossified. It is characterised by a bony depression that occurs behind the coronal sutures.−Sphenocephaly is the most common form and occurs when the middle and posterior third of the sagittal suture are ossified. It is mainly characterised by a bossing of the bregma and of the frontal bone. In these cases, from a bird’s eye view, the width of the frontal bone is greater than the biparietal width.

The frontal and occipital bossing are clinical features of ISS and represent a compensatory phenomenon due to the limited ability of the skull to grow in a lateral direction, according to Virchow’s law. The heterogeneity of the different forms of deformity is, therefore, not only related to the ossification of the sagittal suture per se but, above all, to compensatory mechanisms resulting from the limitation of the normal growth vectors of the skull. It is important to point out that these compensatory processes differ mainly according to the time at which ossification of the sagittal suture takes place in the prenatal period. Thus, early ossification of the suture is associated with a more pronounced scaphocephalic deformity, whereas late ossification of the suture is associated with a less pronounced deformity [12]. These compensatory processes may persist after surgery and may also affect the outcome of surgical treatment, especially if the procedure did not provide sufficient relief or was performed too late.

ISS and the resulting scaphocephaly, with all the morphological forms described above, can be diagnosed at any age. Prenatal diagnosis is quite rare but possible. Prenatal ultrasound measurements in children with ISS are usually within normal limits, although in some cases, a reduced biparietal distance and an increased anteroposterior (fronto-occipital) distance can be observed. Only after birth can the deformity be clearly identified, and the characteristic bony ridge above the sagittal suture can be palpated.

## 3. Radiology

As we have already written, the clinical picture of ISS is not homogeneous. There are different morphological forms of scaphocephalic deformity, and in each of these forms, specific cranial and intracranial changes occur during development. Although CT and MRI are not usually necessary investigations for surgical treatment planning, many surgeons use CT scans for preoperative planning. Both imaging techniques can be useful for research purposes as they can reveal many intracranial abnormalities or variations, not only in relation to the shape of the cranial sulcus but also in relation to the structure of the brain itself. It is clear that changes in the shape of the cranial sulcus can also affect the deformation and shape of the brain itself [13], which becomes longer in ISS compared to normocephalic children. In addition, other deformities are characteristic of the ISS brain, in particular the narrowing of the occipital lobes, the lengthening of the lateral ventricles, and the lateral widening of the frontal lobes. The thalamus and other deep nuclei are slightly displaced backward compared to the brains of children without craniosynostosis. The subarachnoid spaces are distributed differently in children with ISS than under normal conditions. There may be characteristic subarachnoid accumulations or widening of the subarachnoid spaces in the frontal region (Figure 4) and in the interhemispheric fissure. The lateral ventricles may also be slightly wider than normal, especially in the anterior part of the lateral ventricular bodies. These findings are found in up to two-thirds of children with ISS [11].

Two mechanisms have been described that could explain the occurrence of these rearrangements in the intracranial spaces. These are:the above-mentioned passive accumulation of CSF in the subarachnoid spaces as a result of morphological enlargement in the frontal region; andCSF retention as a result of a disturbance in the process of CSF resorption [14].

On the basis of this second theory, the physiopathological process of the development of the bossing of the frontal bone could be at least partially related to the above-mentioned CSF retention. In favour of this hypothesis [14] is the fact that CSF retention is more frequently observed in the frontal subarachnoid cisterns when a bony ridge is seen on CT around the synostotic sagittal suture, surrounding the superior sagittal sinus as a partial or complete bony ring. This is referred to as an Omega sign (Figure 5).

When this groove is absent and the bony surface overlying the superior sagittal sinus is flat, the CSF accumulations mentioned above are less common. In the past, impaired CSF resorption in children with ISS has also been demonstrated by the technique of infusion into the lumbar subarachnoid space [15,16]. These studies have also shown a decompression of the sagittal sinus after surgery. This improved CSF reabsorption and reduced the extent of subarachnoid CSF collections.

In addition to all these changes, some studies have also described changes in intracranial volume in children with ISS based on CT scans. The data in the literature are rather confusing and often contradictory. In fact, studies using CT scans to assess brain volume have shown that intracranial volume in ISS can remain within normal limits in the first months of life despite the deformity and that it increases with age compared to normocephalic children [17,18,19]. Other studies [20] have shown partially different results, namely that the intracranial volume in ISS between the third and tenth months of life is lower than in children of the same age without craniosynostosis [20]. These divergent results show that the natural course and process of cranial growth in children with ISS are complex, only partially understood, and unpredictable.

## 4. Genetics

Although ISS is the most common craniosynostosis, the aetiology of this pathology remains unclear. In recent years, a number of genes have been identified that regulate and control the development and process of progressive, physiological ossification of the sagittal suture. The most important genes involved in this ossification process are FGFR–1–3, TWIST1, RAB23, BMP, EFNB1, TCF12, and PHEX [21]. The diversity of these genes confirms that ISS is also a heterogeneous pathological entity at the molecular level [22]. 

There is no autosomal inheritability for ISS, and the origin of mutations in the aforementioned genes is multifactorial. In this sense, a second pregnancy will very rarely present with the same condition [21]. 

Routine genetic testing in children with ISS has limited indications, and it is questionable whether it is clinically appropriate to perform these tests in all children with this diagnosis or only in children in whom a genetic or metabolic defect is suspected. Otherwise, in most cases where an underlying metabolic disease is present (hypophosphatasia, hypophosphatemic rickets, mucolipidosis, mucopolysaccharidosis, osteopetrosis, pseudohypoparathyroidism), this is usually diagnosed before or at the same time as ISS. More rarely, the reverse is the case, where craniosynostosis is recognised and diagnosed first, which serves as a trigger for further diagnoses leading to the final definition of metabolic disease. There is no doubt that systematic genetic testing, routinely performed in the neonatal period, will make an important contribution to the early detection of metabolic diseases in the coming years, including with regard to the diagnosis of ISS [21,22].

## 5. Intracranial Pressure in ISS

ISS is usually the cause of two different problems: aesthetic deformities andthe risk of developing increased intracranial pressure. 

The occurrence of increased intracranial pressure with ISS is a complication that is usually due to delayed surgery or poor follow-up of the child after surgery. In rare cases, this complication is attributed to a lack of knowledge about the pathophysiology of ISS itself and limited diagnostic capabilities. Knowledge of the correct management, treatment, and choice of surgical technique is also limited. Opinions still differ in the literature in this regard. Therefore, there are no clear guidelines for the ideal treatment of ISS [23].

The possibility of increased intracranial pressure varies widely, as do the consequences of this condition in children with ISS. In Arnaud’s 1995 study, intracranial pressure values measured in the epidural space in the head in 142 patients with ISS at one year of age were not consistent [24]. Values between 3 and 25 mmHg were described. Routine measurement of intracranial pressure in children with ISS is rarely performed, and elevated intracranial pressure is usually diagnosed on the basis of clinical and radiological signs. In this context, it is important to emphasise the importance of diagnosing the condition of papilledema and the resulting potential impairment of visual function. Compared to other forms of craniosynostosis, visual impairment due to papilledema is very rare in ISS, but conversely, in cases where ISS is associated with increased intracranial pressure, papilledema and impaired visual function are common findings. The prevalence of papilledema in ISS is estimated to be approximately 5% [25].

## 6. Neurocognitive Development in ISS

Increased intracranial pressure not only leads to visual problems but also to a delay in neurocognitive development. As we have seen, the different forms of scaphocephaly can vary depending on the aetiology, morphology of the cranial vault, age at diagnosis, and intracranial findings on imaging of the head. These different conditions can result in different neurocognitive sequelae, which can vary greatly. In this context, the outcomes in terms of learning ability and neurocognitive function may also vary. A number of pathophysiological factors, alone or in combination, may explain the occurrence of neurocognitive problems associated with ISS:increased intracranial pressure [26],deformation of the brain as a result of deformation of the skull [27],problems with the normal development of the brain,compression of the venous sinuses with resulting impairment of venous outflow [14,28].

The scores of the intelligence quotient (IQ) in these children are usually in the average range [27], and there are even studies in which some children with ISS achieved high or very high IQ scores [29,30]. Nevertheless, neurodevelopment is at least partially impaired in children with ISS, and many authors believe that children with ISS are at increased risk of neurodevelopmental disorders, which manifest primarily in the form of learning difficulties in early childhood and adolescence [9,27]. In addition, several studies have shown that the neurocognitive performance of children who have not operated ISS in intelligence tests was in the average range but lower compared to unaffected children. While motor and cognitive development in the first years of life may be in line with expectations, later performance on developmental tests may be lower than expected [31].

Magge et al. described that 50% of children with ISS between the ages of 6 and 16 had learning and reading difficulties, although all of these children had normal IQ scores on intelligence tests [9]. Other authors have also described poorer gross motor skills and significantly lower non-verbal IQ than verbal IQ in children with ISS compared to controls [29,32]. In another study, children operated on ISS performed better on cognitive organisation scales than the control group, whereas they performed significantly worse on working memory and information processing speed scales [33]. Arnaud [24] demonstrated a negative correlation between IQ scores and high ICP values. Of the children with ISS and elevated ICP, 16% showed a developmental delay, while of the children with ISS and normal ICP, 6% showed a developmental delay. Although the difference was not statistically significant, there was a trend suggesting a negative association between ICP and cognitive development, which has been questioned by several other authors in the past [34].

## 7. The Treatment of ISS

The aim of surgical treatment for sagittal craniosynostosis is to reopen the cranial suture and allow the skull to grow and develop in all directions. This is because the growth of the skull is based on the pressure exerted by the underlying and growing brain, which thus directs and controls the development of the skull. There are differing opinions as to whether the purpose of ISS surgery is purely cosmetic, i.e., aesthetic, or whether it is also necessary to counteract the effects of increased intracranial pressure resulting from the limited capacity of cranial growth and disturbances in physiological CSF circulation [35].

In seeking an answer to these questions, it is worth noting that, unlike most other craniosynostoses, signs of scaphocephaly can also be seen in some cognitively normal adults [11,35]. This is a rare finding but has raised some doubts and questions in the past about the real purpose of surgical treatment, which, based on these facts, should have a predominantly or exclusively aesthetic purpose. Against this background, the early surgical treatment commonly offered to children with ISS cannot be considered absolutely necessary. On the other hand, these doubts are clearly refuted by a number of other studies that clearly show the importance of early surgical treatment. For example, it has been shown that older children diagnosed with ISS after the age of 4 often showed signs of chronically elevated intracranial pressure [11,30].

The two main goals and purposes of surgical treatment are:prevention or treatment of brain dysfunction when clinical signs of increased intracranial pressure are already present,the aesthetic correction of skull deformity.

These two goals are achieved by 

increasing the volume of the skull,redirecting the vectors of cranial growth,normalising the dynamics of the skull,correcting the aesthetic appearance.

## 8. Surgical Techniques

Virchow was the first to propose a modern theory of the pathophysiology of craniosynostosis in 1851 [2]. Subsequently, the French surgeon Lannelongue was the first to describe the surgical treatment of craniosynostosis in 1881 [36]. He described the linear craniectomy, in which a bone ligament was removed parallel to the ossified sagittal suture. Later, in 1882, the English surgeon Lane operated on a nine-month-old child who was diagnosed with craniosynostosis in conjunction with microcephaly [37]. He removed the ossified suture from the anterior to the posterior fontanel and performed a bilateral parietal osteotomy so that the osteotomies formed a cross shape. In 1894, Jacobi [38] described a high morbidity and mortality rate in a series of 33 craniosynostosis patients, which even led to craniosynostosis surgery not being performed for the next three decades. Faber and Towne [39] reintroduced surgical treatment for craniosynostosis patients, mainly to prevent visual impairment and blindness. They emphasised the need for surgical intervention in early childhood, between the ages of one and three months. Since then, many surgical techniques have been developed and described for the treatment of ISS. Historically, the first surgical interventions for ISS were limited to the removal of the ossified suture, known as linear suturectomy. Over time, more and more extensive surgical steps were added to this basic procedure, leading to invasive techniques to reshape the entire cranial vault. Despite the more or less invasive individual surgical techniques, the main purpose of open surgery remains the removal of the ossified suture, to which more or less invasive steps are then added to reshape the skull deformity that constitutes the clinical picture of ISS [40,41,42,43,44,45].

A number of open surgical techniques have been demonstrated and described to actively reshape the convexity of the forehead. All these techniques are based on the removal, displacement, and reimplantation of free bone flaps [41,42,46,47,48,49,50], which allow a more or less pronounced bilateral widening and shortening of the anteroposterior cranial sulcus. In recent decades, this principle has been pursued less through extensive surgical procedures and more through the use of embedding material, usually springs, to reshape the cranial vault. Similarly, endoscopic techniques have gained popularity in the last 10 years and are commonly used for the treatment of ISS. In the endoscopic technique, the endoscope is used to perform a sagittal suturenectomy and short parietal osteotomies. This procedure, therefore, does not aim to radically reshape the entire vault but only to relieve the restrictive effect that ISS has on the growth of the cranial scrotum. Further remodelling of the vault is carried out by a corrective helmet, which is used postoperatively and controls growth based on the physiological growth of the brain and enables remodelling of the skull [51,52].

The aim and objective of all these surgical techniques, both open and endoscopic, are, of course, always the same, namely to allow the brain to develop normally and to achieve an aesthetic state in which it is impossible to see that the child had a skull deformity at any time in the past. The aim is, therefore, to give the child a completely normal appearance. The basic principle of any surgical therapy is to counteract the abnormal longitudinal growth of the skull in favour of transverse growth. There are also different opinions about the ideal age for surgery. The general consensus is that cranial remodelling is more aesthetically successful in children operated on before the age of six months. Neurocognitive studies on operated children have also shown that children operated on before this age have a better outcome than others [30,35,53]. In our institution, for example, we performed a biparietal expansion, namely the Renier H technique (RHT), on 28 consecutive children with ISS between 2015 and 2018. However, at the early postoperative follow-up of these children, we found that the frontal protrusion remained visible and aesthetically disturbing (Figure 6). We then changed the surgical protocol and introduced a more extensive reshaping, which we called total cranial vault reshaping (TCVR), which also included the frontal bone and offered the possibility of a better aesthetic result. In our study [1], the majority of children underwent surgery before the age of 6 months. Such early surgery also allows bone defects and ridges that form in the area of extensive osteotomies to be completely covered and hidden by active overgrowth of bone tissue during the first year of life. If children are operated on later, especially after the first year of life, these bone defects can remain palpable or visible. This is unfavourable, especially if they are located in the forehead area, where they are not covered by the scalp, and thus represent an aesthetic problem [11].

Several surgical techniques are described in the literature, but not all of them achieve or fulfil these goals. In scaphocephaly, it is necessary to release the reduction in intracranial space by the synostotic suture and, depending on the surgical technique used, to remodel the entire connective neurocranium to varying degrees. This allows the brain to grow freely in all directions and shape the skull symmetrically. The ideal age for the operation is between three and six months, as the brain still has sufficient growth potential after the operation to give the skull a normocephalic shape. For this reason, no osteosynthetic material is used in these operations that could impair certain growth vectors of the skull in one way or another [11,23].

Despite decades of continuous research, the ideal surgical treatment has not yet been proven, nor has it been proven that one treatment would later show clear advantages over other techniques in the long-term follow-up of these children [23]. Many surgical techniques have been described. The most commonly used are biparietal reshaping with the RHT [54], total vault reshaping, and endoscopic reshaping [35,54,55,56]. Biparietal remodelling or RHT, is based on the expansion of the parietal part of the skull. Many variations in this procedure have been proposed and described, often named after the shapes of the osteotomies performed during the procedure: Pi technique, T technique, Y technique, inverted Y technique, inverted Pi technique, inverted T technique, double Pi technique, double T technique, and double H technique. Even more invasive and extensive are the techniques for reshaping the entire cranial vault, in which not only osteotomies are created, but the entire vault is gradually reshaped more extensively and more actively (Figure 7). These techniques are based on the removal, displacement, and reimplantation of free bone flaps and are considered to be the most extensive and invasive [54,57,58,59,60]. They all go under the common name of remodelling the entire cranial vault. Despite these many well-known open surgical techniques, in recent years, many centres have focused on the introduction of surgical techniques aimed at reducing the morbidity and overall surgical risk of the procedure. To this end, surgical techniques based on small skin incisions have been developed [51,52,56,61] (Figure 8).

These are mainly endoscopic techniques [55] and techniques based on the use of implantable springs (Figure 9) and distractors [51,52,62,63].

Of these less invasive approaches, endoscopic procedures are the most commonly used. They have the advantage of requiring fewer skin incisions and less blood loss and are therefore considered less invasive and more child-friendly [64,65,66,67,68,69,70]. Endoscopic techniques are based on the principle of a minimally invasive surgical approach in which the endoscope is used to assist in the creation of a sagittal suturectomy and relieve osteotomies through two small skin incisions. These new osteotomies have the functions of newly surgically formed cranial sutures. Ideally, such procedures are performed on children at a young age, between the first and third months of life. It is important that children wear a corrective helmet for several months after such an operation. It is the wearing of the helmet that determines the final success, not the primary operation itself. Endoscopic treatment is based on the ability of the developing brain to reshape the skull, and the corrective helmet provides adequate support by inhibiting the longitudinal growth vector of the skull and thereby stimulating the lateral growth vector [71,72].

Several studies have shown the efficacy of endoscopic treatment of ISS and comparable results to those after open cranial remodelling [71,72,73].

The advantages and limitations of each of these individual surgical techniques have been extensively analysed in numerous articles, mainly based on surgical criteria such as:Blood loss during surgery: The amount of blood loss during endoscopic craniosynostosis surgery can vary depending on factors such as the complexity of the case, the patient’s medical condition, and the surgical technique employed. It is difficult to provide exact quantities as they can differ widely, especially between individual cases. Endoscopic craniosynostosis surgery is typically associated with minimal blood loss compared to traditional open procedures. In many cases, blood loss may be limited to a few millilitres to tens of millilitres.The need for blood transfusions: The need for blood transfusions during endoscopic craniosynostosis surgery varies depending on several factors, including the patient’s age, medical condition, the complexity of the surgery, and the amount of blood loss experienced during the procedure. Paediatric patients have lower blood volumes compared to adults. As a result, even a small amount of blood loss relative to body size can have a more significant impact and may necessitate a blood transfusion. Despite efforts to minimise blood loss, some degree of bleeding can occur during surgery. Patients with pre-existing anaemia or lower-than-normal haemoglobin levels may be at higher risk of requiring blood transfusions during surgery, particularly if significant intraoperative bleeding occurs. Endoscopy is, therefore, a suitable surgical technique for this group of patients. The duration of the operation: The duration of endoscopic craniosynostosis surgery can vary depending on several factors, including the complexity of the case, the specific techniques employed, the number of sutures involved, and the surgeon’s experience. Generally, endoscopic craniosynostosis surgery tends to be shorter in duration compared to traditional open cranial vault reconstruction. The duration of endoscopic surgery typically ranges from 1 to 4 h, depending on various factors, including the number of involved sutures, the experience of surgeons, and surgical technique, among others. The length of hospitalisation.

## 9. Clinical Outcome of the Surgical Treatment

It must be said that long-term results, both aesthetic and functional, are rarely demonstrated in most studies. Therefore, despite the relatively large number of studies, we do not have the data to draw firm conclusions about the long-term effectiveness of certain surgical techniques. As far as the aesthetic outcome is concerned, the conclusions of these studies are mainly based on postoperative measurements of the CI (ratio of skull width to skull length), which is not a perfect indicator of the morphology of the cranial body. At the same time, the results of the various studies are very heterogeneous, as children with different initial stages of deformity, different ages, and different surgical techniques were included in the studies. In any case, all studies describe an improvement in CI after surgery, and it is generally recognised that the lateral, biparietal head circumference increases after the surgery while the anteroposterior head circumference decreases to a lesser extent [74].

Different surgical techniques may have different effects on the final morphological appearance of the skull. In his study, Panchal compared linear suturectomy (without postoperative wearing of a helmet) with whole skull remodelling and described a more significant improvement in CI after whole skull remodelling compared to suturectomy one year after surgery. However, the long-term outcome was not analysed in this study, so no definitive conclusions could be drawn [75].

Although the aesthetic outcome is usually only assessed on the basis of a subjective evaluation, and these results should, therefore, be treated with caution, the literature shows that the aesthetic outcome is rated as good or very good in most cases. Only a few children show a poor aesthetic result and the need for a second operation. In this context, it should be noted that surgical treatment can also lead to more or less large bony defects in the vaulted skull, which occasionally require subsequent surgical coverage or cranioplasty.

## 10. ISS Relapse and Secondary Craniosynostosis of a Different Suture

Regardless of the surgical technique used, ISS recurrence is one of the rare but important causes of poor long-term aesthetic outcomes. In a series of 79 children who underwent linear suturectomy [76], four children required reoperation due to the recurrence of craniosynostosis and the occurrence of increased intracranial pressure. In the second series, which included 181 children operated for ISS, eleven children had to be reoperated due to proven recurrent synostosis [54,57,58,59,60]. Clinical and radiological evidence of increased intracranial pressure was found in six cases [74]. It should certainly be noted that in some of these children, sagittal craniosynostosis was part of the syndromic disease and that the predisposition for the occurrence of increased intracranial pressure was, therefore, increased in these children. This shows, among other things, the importance of the fact that even in apparently simple forms of ISS, there may be an underlying syndromic disease that is not yet recognised in the early neonatal period. On the other hand, these results indicate that the risk of increased intracranial pressure is always present, even in children with non-syndromic ISS. A strict follow-up after surgical treatment permits the early recognition of a persistent scaphocephalic deformation. An ophthalmologic diagnosis of papilledema and a radiological confirmation of a recurrent synostosis are the criteria on which the diagnosis of a recurrent ISS is based [74,75].

Other studies have also confirmed the risk of postoperative increased intracranial pressure and have shown that secondary craniosynostosis of another previously opened cranial suture may occur after surgery. Postoperative synostosis of the coronal suture is the most common in this context and can occur in up to 10% of children after removal of the sagittal suture if both coronal sutures were not also removed at the time of surgery [77]. Secondary synostosis of the coronary or lambdoid sutures may also occur after remodelling of the entire cranial vault, according to the literature [78]. Finally, this condition may also occur in some children who have not undergone surgery for ISS at all [77,79].

## 11. The Effect of Treatment on Neurocognitive Outcome

With appropriate and successful surgical treatment of ISS, the likelihood of neurocognitive impairment in children is very low and has been reported to be up to 9% in a mild form [3]. However, as we have already seen, the narrower shape of the cranial vault can result in the growing brain not having enough space for normal physiological development, which can lead to clinical signs of increased intracranial pressure, i.e., visual disturbances due to papilledema, headaches, and neurocognitive delays [80].

The impact of surgery on long-term neurocognitive functional outcomes is not clearly known, although surgery for scaphocephaly has been performed for several decades. The results in the literature are contradictory, and, most importantly, most studies have significant methodological limitations so that no realistic and long-term conclusions can be drawn. Different studies have compared different initial levels of deformity, different ages at the time of surgery, different surgical techniques, and different ages at clinical outcome assessment. All these factors, especially age at surgery, surgical technique, type of anaesthesia, and duration of surgery, can influence the final clinical outcome in one way or another. Neuroanaesthesia may also pose a risk for further cognitive delays. Taking all these limitations into account, we can conclude that most studies agree that children with ISS can develop long-term neurocognitive problems, particularly with regard to language development, despite surgical treatment. In a population of 18-year-olds operated on before the age of 30 weeks using a linear sagittal suture technique, a delay in language development was demonstrated [81]. Similar language problems were also demonstrated in another study [30] in which language development was delayed in 28 of the 76 children operated on. It has also been shown that children’s motor functions generally improve after surgery and that these improvements persist over time [32].

Age at surgery is obviously important and has an impact on postoperative status and neurocognitive improvement. Several studies have shown that children operated on in the first year or first six months of life have better outcomes than children operated on later or those operated on at a later age for objective reasons (poor general condition or late referral to a surgical facility). Indeed, children may be referred for surgery because of a late diagnosis of ISS at the time of hospitalisation due to a general neurocognitive delay [1,30,32].

In the study by Hashim [35], the long-term neuropsychological outcome of children operated on for ISS was compared by age at surgery and by the surgical technique used. He compared linear suturectomy (without postoperative wearing of a helmet) with open total cranial vault remodelling. Neuropsychological tests were performed on 70 children (age range 5.75–24.42 years—mean age 10.04 years). The study included measures of IQ, verbal ability, reading ability, and reading comprehension. The worst results were found in the children for whom the intervention was carried out after the age of 12 months. However, in the two groups of children operated on before six months of age, linear suturectomy was associated with worse outcomes (i.e., it was not possible to expand and remodel the cranial vault sufficiently) than the remodelling of the entire vault. This showed that the surgical technique also has an impact on the final neurocognitive outcome and that the functional outcome varies depending on the surgical technique used and the age at surgery. Considering all these data, it is worth considering that the choice of surgical technique has an impact not only on the aesthetic but also on the functional outcome of children with ISS [35].

## 12. Conclusions

ISS continues to be a fascinating field of research that combines genetics, radiology, embryology, and clinical medicine. Advances in diagnostic methods and surgical techniques continue to improve our understanding and patient outcomes. The long-term outcomes of intervention for ISS are multifaceted and include not only the physical aspects of skull morphology but also cognitive development and psychosocial well-being. Early intervention has been associated with better neurocognitive outcomes, emphasising the importance of timely diagnosis and treatment. The importance of interdisciplinary collaboration in the comprehensive management of ISS and the need for continued research to further unravel the complexity of this anomaly cannot be overemphasised.

## Figures and Tables

**Figure 1 diagnostics-14-00435-f001:**
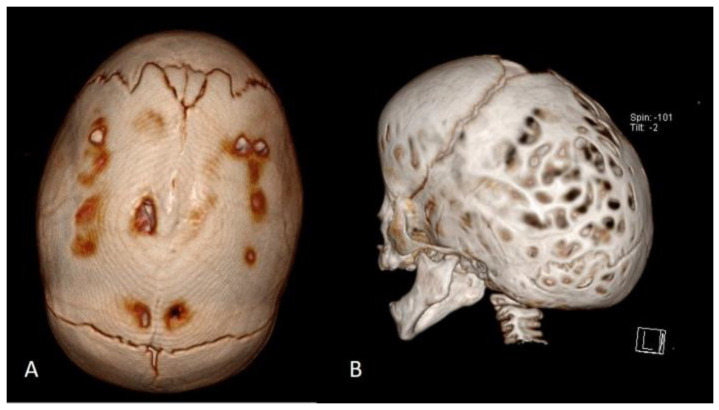
Computed tomography (CT) with 3D reconstructions showing sagittal craniosynostosis (**A**) and subsequent scaphocephalic skull deformity (**B**).

**Figure 2 diagnostics-14-00435-f002:**
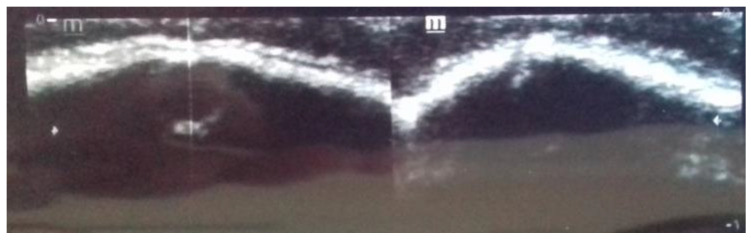
Cranial bone ultrasound confirms the synostosis of the cranial suture, as the hyperechoic bone signal continues uninterrupted, whereas a hypoechoic cranial suture should be identifiable.

**Figure 3 diagnostics-14-00435-f003:**
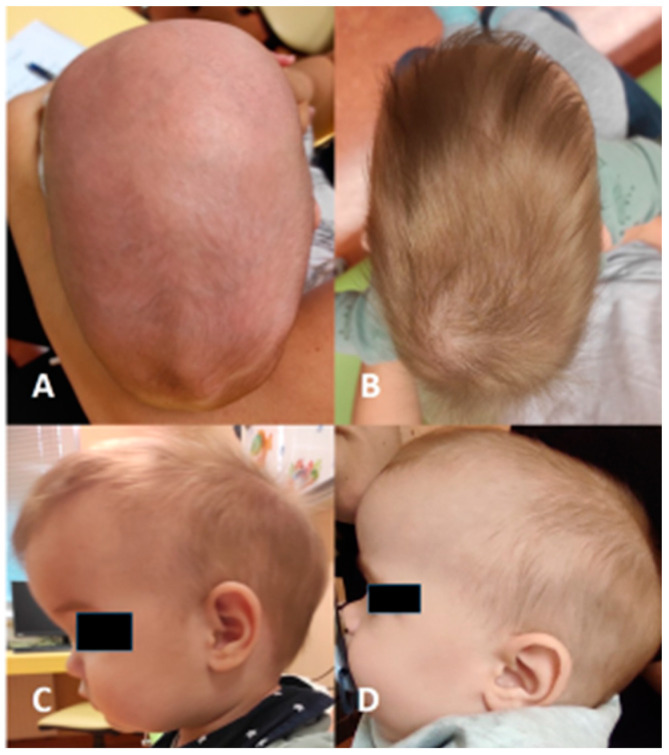
(**A**,**B**) show the scaphocephalic head shape from a bird’s eye view, characterised by biparietal constriction and increased anteroposterior circumference. (**C**,**D**) show the lateral view, with prominent frontal convexity and flattening of the vertex.

**Figure 4 diagnostics-14-00435-f004:**
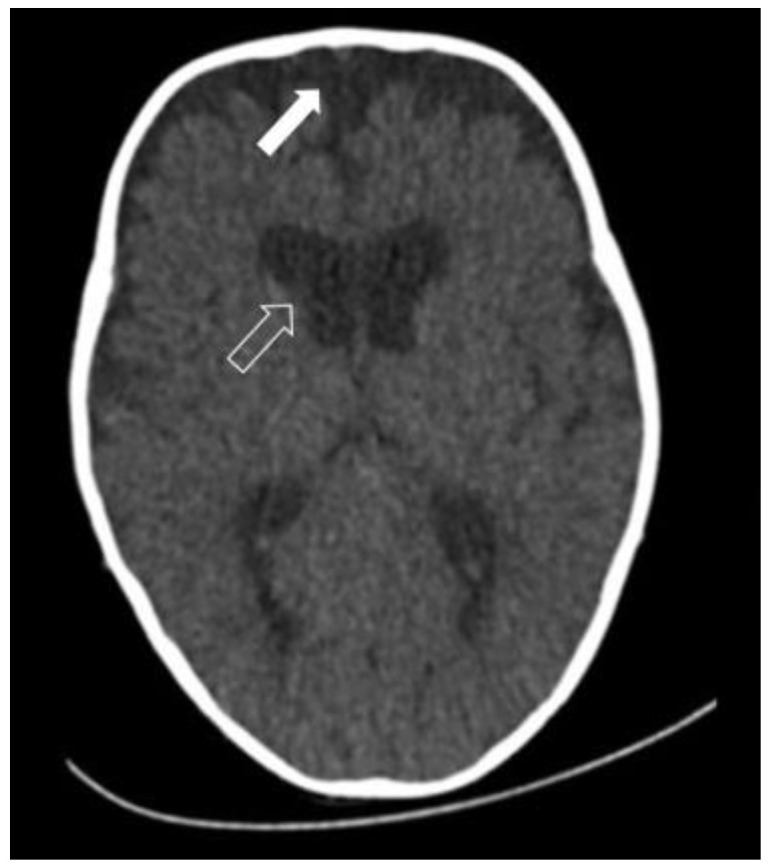
A CT scan of scaphocephaly showed enlarged frontal pericerebral subarachnoid spaces (white arrow) and slightly wider frontal horns of the lateral ventricles (empty arrow).

**Figure 5 diagnostics-14-00435-f005:**
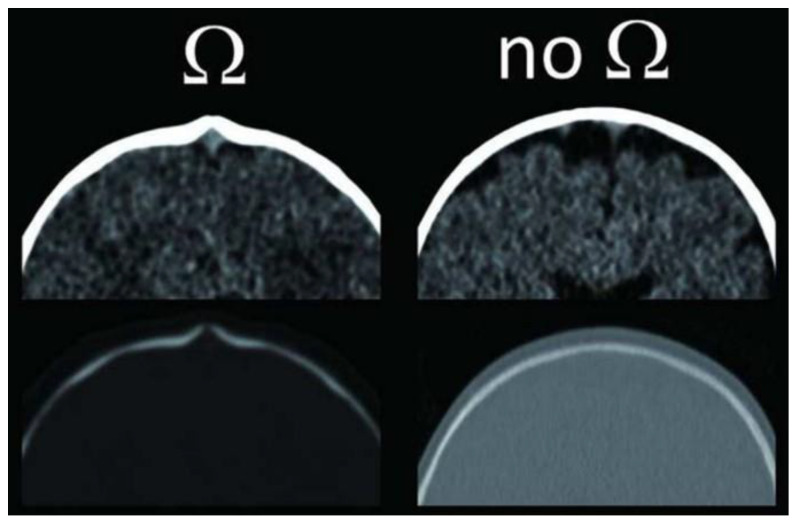
The Omega sign is recognisable on CT scans. It is a bony groove in the area of the synostotic sagittal suture, which hugs the superior sagittal sinus (refer to Ref. [14].)

**Figure 6 diagnostics-14-00435-f006:**
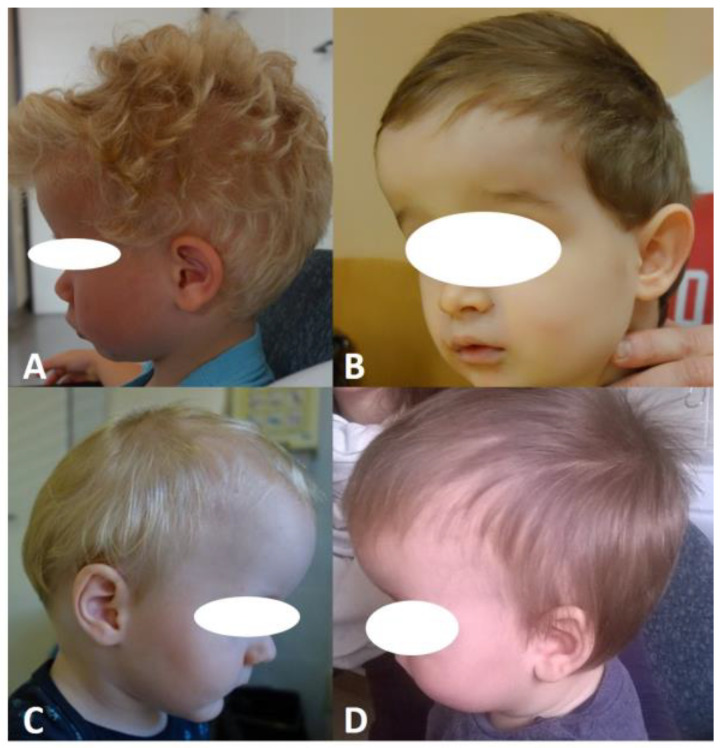
Four cases of children (**A**–**D**) with marked and persistent frontal bulging at early follow-up after RHT surgery.

**Figure 7 diagnostics-14-00435-f007:**
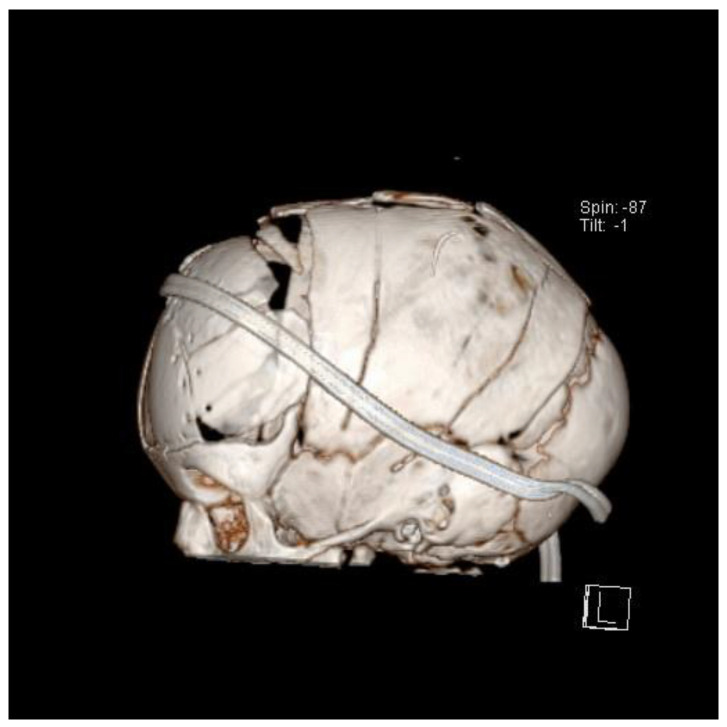
The reshaping of the entire vault involves not only biparietal expansion but also the reshaping of the frontal bone.

**Figure 8 diagnostics-14-00435-f008:**
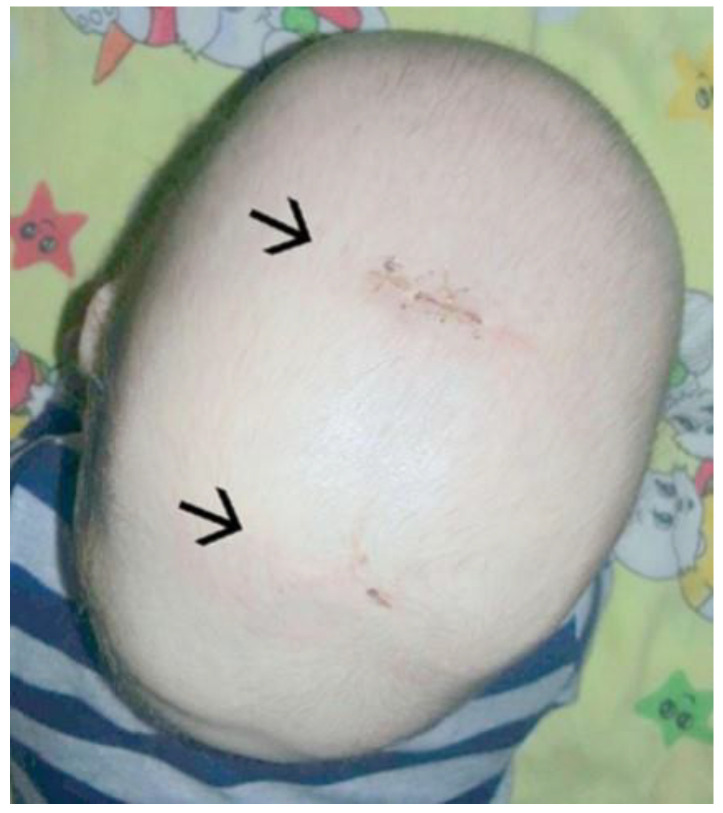
The aim of an endoscopic procedure is to minimise the invasiveness of the surgery itself. The endoscopic approach is performed through two small incisions in the skin (arrows). The synostotic sagittal suture is then removed with the assistance of an endoscope.

**Figure 9 diagnostics-14-00435-f009:**
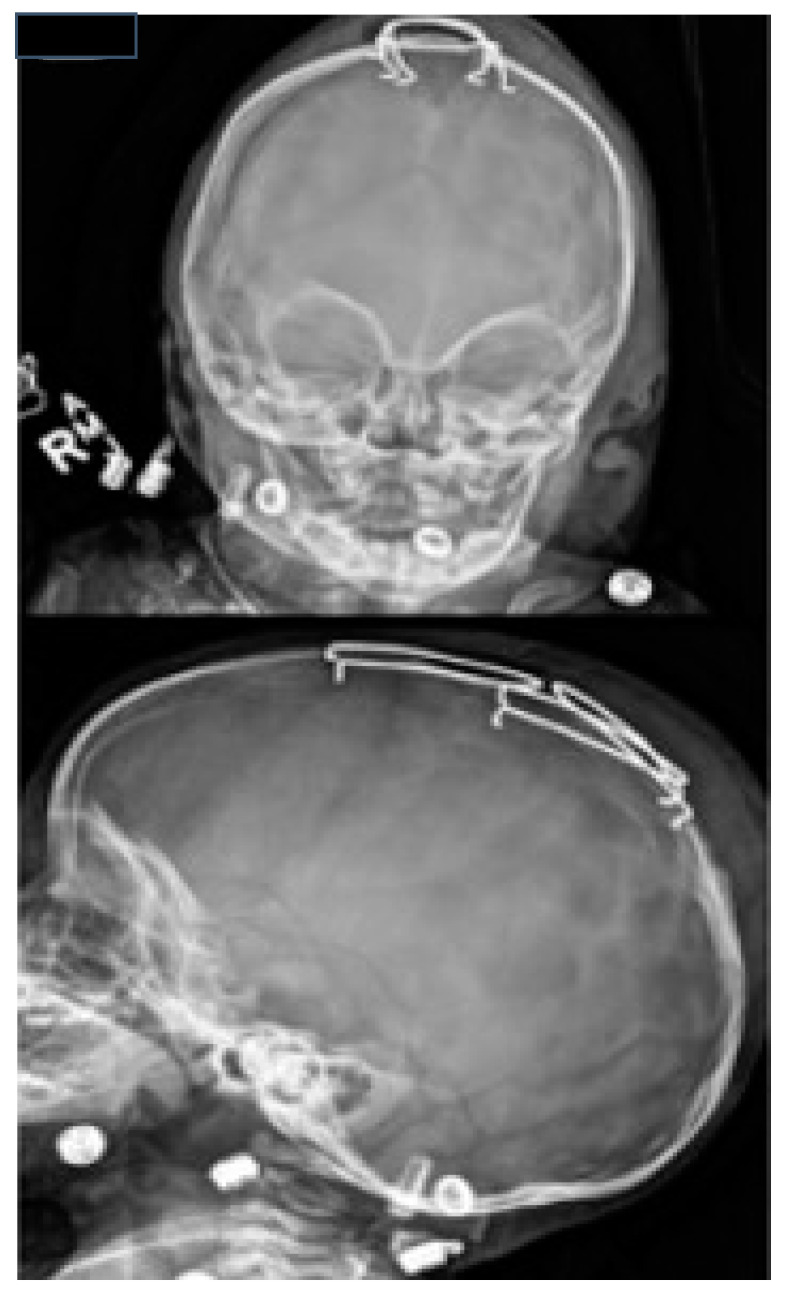
One of the modern surgical methods of ISS treatment involves the insertion of springs into the bone defect created after the suturectomy (refer to Ref. [62]).

## Data Availability

Not applicable.

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
