# Peer review of "Isolated Sagittal Craniosynostosis: A Comprehensive Review"

_diagnostics, 2024, doi:10.3390/diagnostics14040435_

Round 1

Reviewer 1 Report

Comments and Suggestions for Authors

Dr. Spazzapan and Velnar provide a comprehensive review of the diagnosis, surgical treatment, and outcomes of sagittal craniosynostosis. Provided below are comments listed by line number. Many of the comments are related to word choice and potentially translation errors.

 38- It appears that the authors meant to say “base of the skull”

 48- (and elsewhere) “fused” should replace “atrophied”

 55- Many syndromic forms of craniosynostosis are not associated with hydrocephalus or Chiari malformation (e.g. Muenke syndrome) and are relatively uncommon in Apert syndrome relative to Crouzon syndrome. The description “usually associated” is misleading. This reviewer does not suggest that you necessarily expand on this description but rather give a more general statement such as “Syndromic forms of craniosynostosis usually present with multiple fused sutures and may have CNS, limb, or airway malformations”.

 56- It is unclear what is meant by “interlobar hypertonia” – do the authors mean increased intracranial pressure?

 77- This reviewer would suggest removing the word “rarely”. Microcephaly due to perinatal injuries often results in craniosynostosis.

 81-  Mutations in TGFs do not cause craniosynostosis in humans. It is assumed the authors meant TGFBR2 associated with Loeys-Dietz syndrome. If the authors wish to keep the sentence listing genetic causes of craniosynostosis this reviewer would suggest including TWIST1 and ERF. Others such as EFNB1 could also be included. It is important to stress that this section is referring to syndromic forms of craniosynostosis.

 85- Did the authors mean “calvarium” rather than “capitellum”?

 97- MRI imaging is frequently used in syndromic craniosynostosis to evaluate for Chiari malformation, syrinx, hydrocephalus, and other CNS malformations.

 106- It is unclear what is meant by “surgical exposure”.

 118- “Time consuming” makes the treatment sound burdensome. Consider saying “…complex and requires multidisciplinary care.”

 122- Liquorrhhea is not a commonly used term. Did the authors intend to describe nasal CSF drainage?

 129- (paragraph beginning at 129) – This paragraph appears to be describing neurocognitive problems in syndromic craniosynostosis. If that is not the case neurocognitive problems in 30-50% of cases with sagittal synostosis seems much to high.

 165-177 – It is unclear why descriptions of all these rarely used terms adds to the manuscript.

 175- Sternum?

 186- Scrotal?

 199- Many teams use CT scan for preoperative planning.

 Figure 4 – It should be stated that these are CT scans from two individuals. Also, because CT scans are obtained with the patient I a supine position, there always appears to be more extra axial CSF space in the frontal area.

 221-223- These statements need references.

 236- here and elsewhere “lytic accumulations” should just be described as CSF. Unless I misunderstood the purpose of the sentence.

 256- It appears that the authors meant TWIST1. This author would suggest listing only those genes commonly associated with syndromic craniosynostosis. It seems that this list should match the previous list.

 260- and elsewhere – it is unclear what is meant by “metabolic defect?

 269- Is intraluminal pressure equivalent to intracranial pressure. If so, intracranial pressure is a much more commonly used term. If not, it should be defined.

 277- Were the 142 patients evaluated with intracranial pressure monitoring or lumbar puncture?

 279-284- It is unclear what “intraocular pressure”, “optic disc stenosis”, and “occluded optic disc” refer to. Fundoscopic examination looking for evidence of papilledema is the standard. If the terms used are specific evaluations or observations, they should be defined.

 304- “Nevertheless…..with ISS” reads as though all children with ISS have neurocognitive impairment. This sentence should be revised.

 308- How were patients defined as being inoperable?

 312-323- This paragraph contains a lot of information. Consider synthesizing the information. The authors might consider using a table to show these data.

 349- What is meant by normalization of the dynamics of the skull?

 358- Did the authors intend to say “posterior scapula”

 378- “….to reshape the spinal cord”. Did the authors mean spinal cord?

 384-385- “scrotum” appears to the wrong word.

 392- This reviewer would suggest saying theta there are “differing opinions” rather than “confusion”

 400- Did the authors intend to say “acetabular” reshaping?

 387- This paragraph stands out in the paper because it includes the authors opinions about care and the rest of the article references published works. This reviewer would suggest that the authors only include information from their publications in this section.

 413- what is meant by “constricting difficulty”

 416- What does it mean to say the brain is allowed to grow “physiologically”

 451- Unless this is by request of the Journal the reference cited in figure 9 should just be numbered like all of the other references.

 458- Explain “child friendly”

 461- Reword “…..two small incisions that assume the role of physiological cranial suture postoperatively”. Certainly the incisions have nothing to do with the sutures.

 464- Headgear should be “helmet”

 473-475- The authors should specify less blood loss, less frequent need for transfusion, and shorter duration of surgery. As written the reader would not understand the benefits.

 502- Consider using the term “relapse” instead of recurrence which sounds like genetic recurrence.

 506- Was “intraluminal” intended to be intracranial?

 509- It was surprising that the authors would include data on sagittal synostosis as part of a syndrome. Patient with syndromic craniosynostosis are much more likely to have increased intracranial pressure.

 515- Did the authors mean “intraocular pressure” or “papilledema”?

 541- It would be good to state tat other factors (e.g., anesthesia) are risk factors for neurocognitive delays.

 550- “…first year or first six months”. Are the authors reference two studies? Otherwise it would seem that only one timeframe needs to be listed.

 557- To my knowledge “suturation” is not a word.

 558- I believe the authors meant to say “barrel stave” procedure. I do not believe that this was described earlier in the text. It should be defined.

 563- What specifically is meant by “worse outcomes”

Comments on the Quality of English Language

Included in my review.

Author Response

Reviewer 1

Thank you for the review and comprehensive suggestions. We have corrected them directly into the text (changes and corrections are marked in red) and we apologise for the wrong words. The text is corrected now and suggested parts have been added. Please find attached short answers, for convenience, also here, below the questions. 

COMMENTS. Dr. Spazzapan and Velnar provide a comprehensive review of the diagnosis, surgical treatment, and outcomes of sagittal craniosynostosis. Provided below are comments listed by line number. Many of the comments are related to word choice and potentially translation errors.

 38- It appears that the authors meant to say “base of the skull” 

This was corrected in the text.

 48- (and elsewhere) “fused” should replace “atrophied” 

Thank you. Replaced.

 55- Many syndromic forms of craniosynostosis are not associated with hydrocephalus or Chiari malformation (e.g. Muenke syndrome) and are relatively uncommon in Apert syndrome relative to Crouzon syndrome. The description “usually associated” is misleading. This reviewer does not suggest that you necessarily expand on this description but rather give a more general statement such as “Syndromic forms of craniosynostosis usually present with multiple fused sutures and may have CNS, limb, or airway malformations”.

Added and corrected.

56- It is unclear what is meant by “interlobar hypertonia” – do the authors mean increased intracranial pressure? 

We meant intracranial hypertension- increased intracranial pressure.

 77- This reviewer would suggest removing the word “rarely”. Microcephaly due to perinatal injuries often results in craniosynostosis. 

Corrected.

 81-  Mutations in TGFs do not cause craniosynostosis in humans. It is assumed the authors meant TGFBR2 associated with Loeys-Dietz syndrome. If the authors wish to keep the sentence listing genetic causes of craniosynostosis this reviewer would suggest including TWIST1 and ERF. Others such as EFNB1 could also be included. It is important to stress that this section is referring to syndromic forms of craniosynostosis. 

Corrected and added in red. 

 85- Did the authors mean “calvarium” rather than “capitellum”? 

Cranial vault. Corrected.

 97- MRI imaging is frequently used in syndromic craniosynostosis to evaluate for Chiari malformation, syrinx, hydrocephalus, and other CNS malformations. 

Added, thank you.

 106- It is unclear what is meant by “surgical exposure”. 

Surgical treatment is the correct word.

 118- “Time consuming” makes the treatment sound burdensome. Consider saying “…complex and requires multidisciplinary care.”

Corrected.

 122- Liquorrhhea is not a commonly used term. Did the authors intend to describe nasal CSF drainage?

CSF leakage.

 129- (paragraph beginning at 129) – This paragraph appears to be describing neurocognitive problems in syndromic craniosynostosis. If that is not the case, neurocognitive problems in 30-50% of cases with sagittal synostosis seems much too high. 

We found this data in the literature. Magge SN, Westerveld M, Pruzinsky T, Persing JA. Long-term neuropsychological effects of sagittal craniosynostosis on child development. J Craniofac Surg 2002; 13:99−104.

165-177 – It is unclear why descriptions of all these rarely used terms adds to the manuscript.

Just added form completeness. If the respected reviewer wishes, we can delete them.

 175- Sternum? 

Corrected.

 186- Scrotal? 

Corrected.

 199- Many teams use CT scan for preoperative planning. 

Corrected and added.

 Figure 4 – It should be stated that these are CT scans from two individuals. Also, because CT scans are obtained with the patient I in supine position, there always appears to be more extra axial CSF space in the frontal area.

 Corrected.

 221-223- These statements need references. 

Added.

 236- here and elsewhere “lytic accumulations” should just be described as CSF. Unless I misunderstood the purpose of the sentence. 

Corrected.

 256- It appears that the authors meant TWIST1. This author would suggest listing only those genes commonly associated with syndromic craniosynostosis. It seems that this list should match the previous list. 

Thank you. Corrected.

 260- and elsewhere – it is unclear what is meant by “metabolic defect? 

Metabolic disease. Corrected.

 269- Is intraluminal pressure equivalent to intracranial pressure. If so, intracranial pressure is a much more commonly used term. If not, it should be defined. 

Intracranial pressure. Corrected.

 277- Were the 142 patients evaluated with intracranial pressure monitoring or lumbar puncture?

The measurements were done by epidural sensor inserted in the head. 

 279-284- It is unclear what “intraocular pressure”, “optic disc stenosis”, and “occluded optic disc” refer to. Fundoscopic examination looking for evidence of papilledema is the standard. If the terms used are specific evaluations or observations, they should be defined. 

Corrected and clarified.

 304- “Nevertheless…..with ISS” reads as though all children with ISS have neurocognitive impairment. This sentence should be revised.

Revised.

308- How were patients defined as being inoperable? 

Here, we meant the patients that were not operated on. Apologies for mistake.

 349- What is meant by normalization of the dynamics of the skull?

We have corrected this.

 358- Did the authors intend to say “posterior scapula” 

We meant fontanel.

 378- “….to reshape the spinal cord”. Did the authors mean spinal cord? 

Mistake. Cranial vault meant.

 384-385- “scrotum” appears to the wrong word. 

Corrected.

 392- This reviewer would suggest saying that there are “differing opinions” rather than “confusion”

Corrected.

 400- Did the authors intend to say “acetabular” reshaping? 

Wrong word. Corrected.

 387- This paragraph stands out in the paper because it includes the authors opinions about care and the rest of the article references published works. This reviewer would suggest that the authors only include information from their publications in this section. 

Cited. 

 413- what is meant by “constricting difficulty”

We meant a reduction of the intracranial space. Corrected.

   451- Unless this is by request of the Journal the reference cited in figure 9 should just be numbered like all of the other references. 

Corrected.

461- Reword “…..two small incisions that assume the role of physiological cranial suture postoperatively”. Certainly the incisions have nothing to do with the sutures.

These new osteotomies have the functions of newly, surgically formed cranial sutures.

 464- Headgear should be “helmet”

Corrected.

 473-475- The authors should specify less blood loss, less frequent need for transfusion, and shorter duration of surgery. As written, the reader would not understand the benefits.

Added.

 502- Consider using the term “relapse” instead of recurrence which sounds like genetic recurrence.

Corrected.

506- Was “intraluminal” intended to be intracranial?

Intracranial. Corrected.

 509- It was surprising that the authors would include data on sagittal synostosis as part of a syndrome. Patient with syndromic craniosynostosis are much more likely to have increased intracranial pressure.

Rewritten.

 515- Did the authors mean “intraocular pressure” or “papilledema”?

We meant papilledema. Corrected.

 541- It would be good to state that other factors (e.g., anesthesia) are risk factors for neurocognitive delays.

Added.

550- “…first year or first six months”. Are the authors reference two studies? Otherwise it would seem that only one timeframe needs to be listed.

Added.

 557- To my knowledge “suturation” is not a word.

Corrected.

 558- I believe the authors meant to say “barrel stave” procedure. I do not believe that this was described earlier in the text. It should be defined.

Corrected. We meant total cranial vault remodelling.

 563- What specifically is meant by “worse outcomes”

Corrected.

ANSWER 1. Thank you for the review and comprehensive suggestions. We have corrected them directly into the text (changes and corrections are marked in red) and we apologise for the wrong words. The text is corrected now and the suggested parts have been added.  

Reviewer 2 Report

Comments and Suggestions for Authors

The authors have provided a concise review on the diagnosis, management and outcomes of isolated sagittal craniosynostosis. The article mostly focuses on the surgical management and outcomes of such procedures, which seems appropriate given the nature of the special issue of the journal. 

Some suggestions for the improvement of this manuscript:

Title:

I believe the title is misleading, the authors do not “unveil the enigma”, instead they present it, nor is this review comprehensive as I have identified much missing information.

Lines 38-43

These sentences do not make sense and should be reworded.

Line 186 and elsewhere

The authors use the phrase “scrotal deformity”, I have not come across the work scrotum unless it is referring to the testicles. I am not sure if this is just something I am unaware of in its relation to the brain or if there was a mix-up in translation.

Figure 4

Arrows would be useful to point out the brain structures mentioned in the annotation.

Radiology

In this section the authors outline a number of brain anomalies that can be visulised in individuals with sagittal craniosynostosis. It would be useful, in terms of clinical management of this condition, to add more information (perhaps in the neurocognitive development section) regarding the clinical consequences of this wide range of brain anomalies.

Genetics

This section is incredibly brief however could be extremely important to the diagnosis and management of sagittal craniosynostosis. Genetics is becoming a routine diagnostic tool in numerous conditions and I believe this section needs expanding to cover rare disease that manifests craniosynostosis, for example Apert syndrome and Crouzon syndrome, the inheritability of sagittal craniosynostosis and the GWAS studies undertaken on the condition. A brief paragraph on the development of the condition would also be worthwhile given that the authors conclude that sagittal craniosynostosis can educate us in the field of embryology.

Line 268

A list is given with the numerals (I) and (I)

Line 319

ICP is not defined

Surgical techniques

This section is the most comprehensive part of this review and obviously the authors’ area of expertise. This section would be enhanced by the inclusion of some indications as to when either type of surgery is beneficial and some clinical recommendations based on the authors’ professional experience with the surgical management of this disorder. The authors’ also mention the potential for reoccurrence, but make no clinical suggestions as to how a reoccurrence could be dealt with.  

Comments on the Quality of English Language

Any issues with the English language have been outlined above

Author Response

Reviewer 2

The authors have provided a concise review on the diagnosis, management and outcomes of isolated sagittal craniosynostosis. The article mostly focuses on the surgical management and outcomes of such procedures, which seems appropriate given the nature of the special issue of the journal. Some suggestions for the improvement of this manuscript:

QUESTION 1. Title: I believe the title is misleading, the authors do not “unveil the enigma”, instead they present it, nor is this review comprehensive as I have identified much missing information.

ANSWER 1. We agree that sagittal craniosynostosis still represents a not-completely-known pathological entity. Many aspects remain unknown. In such conditions, it is practically impossible to “unveil” the enigma that it represents. Our title had the aim of attracting the readership and potentially acting as a "fascinating" title. We thought that being the Manuscript a Review, this would be acceptable. If these aims are not reached, then of course, we agree that a more linear and less misleading title is more appropriate. We suggest a possible new Title: “Isolated sagittal craniosynostosis: a comprehensive review”.

QUESTION 2. Lines 38-43. These sentences do not make sense and should be reworded.

ANSWER 2. We corrected and edited the selected part of the Manuscript. Please find here the edited text:

Based on the maturation process, the skull is divided into two main parts: the viscerocranium and the neurocranium. The neurocranium is further separated into the membranous and the cartilaginous neurocranium. The membranous neurocranium undergoes intramembranous ossification and creates the frontal bone, the squamous portion of the temporal bone, the intraparietal portion of the occipital bone and the parietal bone. The cartilaginous neurocranium ossifies via endochondral means and gives rise to the ethmoid and sphenoid bones as well as the petrous and mastoid portion of the temporal bone and the occipital bone. The junctions of the skull bones are called the skull sutures, where the bone tissue is gradually deposited. The sutures therefore act as ossification centres of the skull (1). The skull sutures allow a smooth growth of the brain and a progressive bone deposition and contemporary, an effective mechanical protection of the brain.

QUESTION 3. Line 186 and elsewhere. The authors use the phrase “scrotal deformity”, I have not come across the work scrotum unless it is referring to the testicles. I am not sure if this is just something I am unaware of in its relation to the brain or if there was a mix-up in translation. 

ANSWER 3. We suggest simply to cancel the definition “scrotal”, which, we agree, is not related to the context of cranial deformations. We corrected this in the manuscript. 

QUESTION 4. Figure 4. Arrows would be useful to point out the brain structures mentioned in the annotation.

ANSWER 4. We added the arrows to point the specific CT characteristics related to sagittal craniosynostosis.

QUESTION 5. Radiology. In this section the authors outline a number of brain anomalies that can be visualised in individuals with sagittal craniosynostosis. It would be useful, in terms of clinical management of this condition, to add more information (perhaps in the neurocognitive development section) regarding the clinical consequences of this wide range of brain anomalies.

ANSWER 5. Thank you for the suggestion. We modified the “Neurocognitive development in ISS” section and tried to explain how the different radiological and morphological conditions can result in different neurocognitive sequelae, which can not be predicted just on the basis of specific radiological conditions.

QUESTION 6. Genetics. This section is incredibly brief however could be extremely important to the diagnosis and management of sagittal craniosynostosis. Genetics is becoming a routine diagnostic tool in numerous conditions and I believe this section needs expanding to cover rare disease that manifests craniosynostosis, for example Apert syndrome and Crouzon syndrome, the inheritability of sagittal craniosynostosis and the GWAS studies undertaken on the condition. A brief paragraph on the development of the condition would also be worthwhile given that the authors conclude that sagittal craniosynostosis can educate us in the field of embryology.

ANSWER 6. ISS is not a disease with autosomal inheritability, but the so far recognized genetic mutations are rather multifactorial. We added to the text this important concept, and the fact that a second pregnancy is not at higher risk for this same pathologic condition. 

Regarding the GWAS studies in ISS there is very little literature and the evidence in this field is very weak. More GWAS studies have concentrated on syndromic forms of craniosynostosis, like Apert and Crouzon syndrome, which is however completely different from the ISS topic, to which the Manuscript refers. We do believe, as neurosurgeons, that a discussion about such a complex and not well defined genetic topic, would not increase the quality of this Manuscript. 

QUESTION 7. Line 319. ICP is not defined.

ANSWER 7.

LINE 268. We corrected the list, using a “list format”.

LINE 319. The definition of “high” has been added to the text.

QUESTION 8.  Surgical techniques. This section is the most comprehensive part of this review and obviously the authors’ area of expertise. This section would be enhanced by the inclusion of some indications as to when either type of surgery is beneficial and some clinical recommendations based on the authors’ professional experience with the surgical management of this disorder. The authors’ also mention the potential for reoccurrence, but make no clinical suggestions as to how a reoccurrence could be dealt with. 

ANSWER 8. We have added a clear definition of how in clinical practise a recurrent ISS should be diagnosed. We also tried, through several correction of the text, to make clearer, which are the evidence-based facts that represent the basis for treatment of ISS. 

Round 2

Reviewer 1 Report

Comments and Suggestions for Authors

49 – it is unclear what is meant by “a smooth growth of the brain”

58 – This should say “depending on which suture is fused”

66 – remove the word “other”

72 – This should read “the most common form of craniosynostosis  is non-syndromic  isolated…”

91-115 – The manuscript has much too much emphasis on genetic (syndromic) forms of synostosis. While it would be fine to touch on this information, the manuscript is about sagittal synostosis so this information is not needed in this detail.

123 – Figure 2 appears to have been made of two images. There is an obvious line between the two images and the anatomy does not line up.

135 – delete “and very welcome”

149&167 – “raised”

171 – delete “the”

178 – should read “frontal and occipital bossing”

209 – should be Bathrocephaly

211 – should this read “cynosephaly”.

217 – bossing

240 – “authors” should be “surgeons”

262 – The purpose of figure 4 is unclear. As demonstrated in this figure, there are not consistent brain/CSF finding in sagittal synostosis. The figure on the right does not depict enlargement of the anterior portion if the lateral ventricles.

300 – The genes listed have not been associated with ISS. In fact, sagittal synostosis is infrequently seen in syndromic synostosis. Again, as stated above, it is unclear what the purpose is to discuss genes associated with syndromic synostosis, the manuscript is about sagittal synostosis.

312 – Mucopolysaccharidosis is misspelled.

343 – substitute approximately for “around”

346 – delete “The”

346-354 – this paragraph contradicts itself. Saying both that shape predicts outcome and that shape cannot predict outcome. This paragraph should be clarified.

453 – should read “There are differing opinions”

483 – This sentence says the same thing in two different ways. Effectively that no surgical method has been found to improve short and long term outcomes.

511 – Are there implants used other than springs?

626 – deleted “Sometimes”

Comments on the Quality of English Language

Word choice and sentence structure need to be carefully reviewed.

Author Response

Dear Sir or Madam,

Thank you again for your review and your comments. We have revised the questions raised and have changed and added the relevant text. The issues we addressed are indicated in red. Please find the answers to your questions also below. We hope that the revisions are now satisfactory for publication in Diagnostics.

Reviewer 1

QUESTIONS.

9 – it is unclear what is meant by “a smooth growth of the brain”

This was awkwardly written, we agree. We have corrected it.

58 – This should say “depending on which suture is fused”

Corrected.

66 – remove the word “other”

Corrected.

72 – This should read “the most common form of craniosynostosis  is non-syndromic  isolated…”

Corrected.

91-115 – The manuscript has much too much emphasis on genetic (syndromic) forms of synostosis. While it would be fine to touch on this information, the manuscript is about sagittal synostosis so this information is not needed in this detail.

This was suggested from the second reviewer. We have a little reduced this part. If the respected reviewer agrees, we would like to leave this part as it is, according to the suggestions of the reviewer 2.

123 – Figure 2 appears to have been made of two images. There is an obvious line between the two images and the anatomy does not line up.

This were two patients, hence two figures.

135 – delete “and very welcome”

Corrected.

149&167 – “raised”

Corrected.

171 – delete “the”

Corrected.

178 – should read “frontal and occipital bossing”

Corrected.

209 – should be Bathrocephaly

Corrected.

211 – should this read “cynocephaly”.

Corrected.

217 – bossing

Corrected.

240 – “authors” should be “surgeons”

Corrected.

262 – The purpose of figure 4 is unclear. As demonstrated in this figure, there are not consistent brain/CSF finding in sagittal synostosis. The figure on the right does not depict enlargement of the anterior portion if the lateral ventricles.

We have deleted the second figure.

300 – The genes listed have not been associated with ISS. In fact, sagittal synostosis is infrequently seen in syndromic synostosis. Again, as stated above, it is unclear what the purpose is to discuss genes associated with syndromic synostosis, the manuscript is about sagittal synostosis.

The genetic part was suggested to be broadened by the second reviewer. Genes were also suggested to be added. If the respected reviewer agrees, we would like to leave this part as it is, otherwise, we can delete it.

312 – Mucopolysaccharidosis is misspelled.

Corrected.

343 – substitute approximately for “around”

Corrected.

346 – delete “The”

Corrected.

346-354 – this paragraph contradicts itself. Saying both that shape predicts outcome and that shape cannot predict outcome. This paragraph should be clarified.

453 – should read “There are differing opinions”

Corrected.

483 – This sentence says the same thing in two different ways. Effectively that no surgical method has been found to improve short and long term outcomes.

This part was clarified: A number of pathophysiological factors, alone or in combination, may explain the occurrence of neurocognitive problems associated with ISS …

511 – Are there implants used other than springs?

Thank you. This was corrected: These are mainly endoscopic techniques (55) and techniques based on the use of implantable springs ...

626 – deleted “Sometimes”

Corrected.

Reviewer 2 Report

Comments and Suggestions for Authors

The authors have revised this paper satisfactorily  

Comments on the Quality of English Language

No further corrections needed

Author Response

Reviewer 2

QUESTION 1. The authors have revised this paper satisfactorily  

ANSWER 1. Thank you for this kind comment and the review.